# Sofalcone Suppresses Dengue Virus Replication by Activating Heme Oxygenase-1-Mediated Antiviral Interferon Responses

**DOI:** 10.3390/ijms26135921

**Published:** 2025-06-20

**Authors:** Yu-Lun Ou, Wei-Chun Chen, Chia-Hung Yen, Wangta Liu, Chun-Kuang Lin, Shun-Chieh Yu, Mei-Yueh Lee, Jin-Ching Lee

**Affiliations:** 1Department of Internal Medicine, Kaohsiung Municipal Siaogang Hospital, Kaohsiung Medical University Hospital, Kaohsiung Medical University, Kaohsiung 80756, Taiwan; wsp85148@gmail.com; 2Division of Endocrinology and Metabolism, Department of Internal Medicine, Kaohsiung Medical University Hospital, Kaohsiung Medical University, Kaohsiung 80756, Taiwan; 3Department of Biotechnology, College of Life Science, Kaohsiung Medical University, Kaohsiung 80708, Taiwan; daphny3016@hotmail.com (W.-C.C.); liuwangta@cc.kmu.edu.tw (W.L.); 4Graduate Institute of Natural Products, College of Pharmacy, Kaohsiung Medical University, Kaohsiung 80708, Taiwan; chyen@kmu.edu.tw; 5Department of Marine Biotechnology and Resources, College of Marine Sciences, National Sun Yat-sen University, Kaohsiung 80424, Taiwan; crystalsoul35@gmail.com; 6Department of Biological Sciences, Institute of BioPharmaceutical Sciences, National Sun Yat-sen University, Kaohsiung 80424, Taiwan; b102010022@g-mail.nsysu.edu.tw; 7Division of Endocrinology and Metabolism, Department of Internal Medicine, Kaohsiung Medical University Gangshan Hospital, Kaohsiung 807378, Taiwan; 8Faculty of Medicine, College of Medicine, Kaohsiung Medical University, Kaohsiung 80708, Taiwan; 9Center for Tropical Medicine and Infectious Disease Research, Kaohsiung Medical University, Kaohsiung 80708, Taiwan

**Keywords:** dengue virus, sofalcone, interferon, heme oxygenase-1

## Abstract

Dengue virus (DENV) infection is strongly associated with dengue hemorrhagic fever and dengue shock syndrome, both of which carry mortality risks. Addressing the urgent need for effective dengue therapeutics, we identified sofalcone, a gastroprotective agent with antioxidant and anti-inflammatory properties, as a potential inhibitor of DENV replication. Sofalcone demonstrated efficacy against all four DENV serotypes, with the dose inhibiting 50% (IC50) value of 28.1 ± 0.42 μM against viral replication of DENV serotype 2, without significant cytotoxicity. Additionally, sofalcone significantly improved survival rates and reduced viral titers in DENV-infected ICR-suckling mice. Mechanistically, sofalcone induced heme oxygenase-1 (HO-1) expression via the nuclear factor-erythroid 2-reated factor 2 (Nrf2) pathway, which in turn suppressed viral protease activity and restored antiviral interferon (IFN) responses. This included dose-dependent stimulation of IFN downstream antiviral genes such as 2′-5′-oligoadenylate synthetase 1 (OAS1), OAS2, and OAS3. Given its established clinical use as an anti-gastric ulcer drug, sofalcone offers promising potential for rapid application in treating DENV infection.

## 1. Introduction

The dengue virus (DENV) is an emerging arboviral infection of significant medical concern owing to its substantial disease burden, which ranges from acute, self-limiting febrile illness to life-threatening dengue hemorrhagic fever (DHF) and dengue shock syndrome (DSS) [1]. These conditions can be caused by any of the four antigenically distinct DENV serotypes (1 to 4) [2]. The geographic range of dengue is expanding rapidly, driven by rapid urban development and climate change in tropical and subtropical areas [2]. Approximately 2.5 billion people are at risk of DENV infection, with 400 million cases reported annually [3]. This persistent threat is exacerbated by the absence of effective therapeutic agents and an effective tetravalent dengue vaccine against all four DENV serotypes due to the genetic variability of the RNA virus and the risk of antibody-dependent enhancement (ADE) arising from unbalanced immune responses to multiple serotypes [4], making the discovery of novel therapeutic strategies an urgent priority. DENV, an enveloped virus within the *orthoflavivirus* genus in the *Flaviviridae* family [2], contains an 11 kb RNA genome. This genome encodes a single polyprotein precursor, which is subsequently cleaved by host and viral proteases into ten viral proteins: three structural proteins (C, prM, and E) and seven nonstructural proteins (NS1, NS2A, NS2B, NS3, NS4A, NS4B, and NS5). Among these, NS3 protease, in conjugation with the NS2B cofactor, plays a crucial role on viral replication by exhibiting RNA helicase, serine protease, nucleoside triphosphatase, and 5′-RNA triphosphatase activities. Additionally, it facilitates the evasion of host innate immunity by blocking the stimulator of interferon genes (STING)-mediated antiviral interferon (IFN) responses, further underscoring its importance as a therapeutic agent [5].

Heme oxygenase-1 (HO-1) is an antioxidative enzyme critical for heme catabolism and plays a protective role in various human diseases. It catalyzes the breakdown of heme into biliverdin, carbon monoxide, and free iron, each of which has multiple biological effects, including anti-inflammatory, antioxidative, and anti-apoptotic properties [6]. These functions position HO-1 as a key player in the pathophysiology of neurodegenerative diseases, cardiovascular diseases, cancer, metabolic disorders, and abnormalities in iron metabolism [7]. Following viral infection, the expression or activity of HO-1 increases to safeguard cells from the harmful effects of reactive oxygen species (ROS) generated during viral replication [8]. Previously, we observed that HO-1 expression was lower in DENV-infected patients, which is correlated with the disease severity [9]. The transcription factor nuclear factor-erythroid 2-related factor 2 (Nrf2) regulates HO-1 expression by competing with BTB and CNC homolog 1 (Bach1). Nrf2 upregulates HO-1 by binding to the antioxidant response element (ARE) on its promoter region, while Bach1 acts as a repressor. Under non-stress conditions, Kelch-like ECH-associated protein 1 (Keap1) negatively regulates Nrf2 by promoting its ubiquitination and degradation [10]. Upregulation of Nrf2-mediated HO-1 expression by specific agonists has been linked to antiviral IFN responses against several viruses, including human immunodeficiency virus, influenza A virus, hepatitis C virus, hepatitis B virus, enterovirus, DENV, Zika virus, Ebola virus, herpes simplex virus type 1, and SARS-CoV2 [11]. HO-1 and its metabolites, including ferrous iron, carbon monoxide, and biliverdin, mediate antiviral effects, such as HO-1 interacting with interferon regulatory factor 3 (IRF3) to enhance IFN-β production against Sendai virus infection [12]. Recent studies indicate that HO-1 activation could be an effective strategy for managing COVID-19 infections due to its combined antiviral and antioxidative properties. Additionally, biliverdin, an HO-1 metabolite, suppresses DENV replication by inhibiting DENV NS2B/NS3 protease activity, leading to viral life cycle disruption and subsequent restoration or enhancement of antiviral interferon (IFN) responses [13]. This finding highlights the potential therapeutic value of biliverdin or HO-1 induction in treating DENV infections.

Sofalcone, a derivative compound identified as 20-carboxymethoxy-4,4′-bis(3-methyl-2-butenyloxy) chalcone, was isolated from the root of the Chinese plant *Sophora subprostrata* [14]. Clinically, sofalcone is used as a gastric mucosa protective agent due to its ability to upregulate Nrf2-mediated HO-1 expression [15], suggesting that sofalcone may provide a novel therapeutic strategy by inhibiting DENV replication. By leveraging the Nrf-2-HO-1 signaling cascade, sofalcone can inhibit DENV NS2B/NS3 protease activity, potentially restoring or amplifying antiviral IFN responses and synergistically suppressing viral replication. To further explore sofalcone’s antiviral potential, we extended our research to evaluate its protective effects *in vivo* using the DENV-infected ICR-suckling mouse model, an immunocompetent outbred ICR mice strain [16]. This aimed to assess the compound’s efficacy in the mouse model and elucidate its therapeutic implications for managing DENV infections.

## 2. Results

### 2.1. Sofalcone Suppresses DENV Replication in a Cell-Based and ICR Suckling Mouse Model

To determine the antiviral activity of sofalcone, we first performed a cell-based DENV infectious system to verify the optimal safe and effective dose. DENV-infected Huh-7 cells were treated with varying concentrations of sofalcone, and viral replication was assessed after 3 days by measuring viral protein and RNA levels using Western blotting and RT-qPCR, respectively. Cell viability was measured using the MTS assay. The results demonstrated that sofalcone reduced DENV protein (Figure 1A) and RNA (Figure 1B) levels in a dose-dependent manner, with an IC_50_ value of 10 ± 2 μM, as determined by RT-qPCR analysis, and no significant cytotoxicity was observed at the effective doses against DENV infection. Cell-based immunofluorescence staining further confirmed the antiviral activity of sofalcone against viral infection (Figure 1C). Sofalcone exhibited potent, broad-spectrum inhibitory effects against all four serotypes of DENV, as determined by RT-qPCR (Figure 1D). We subsequently employed a DENV-infected ICR suckling mouse model to assess the antiviral activity and protective effects of sofalcone against lethal viral infection *in vivo*. Six-day-old ICR suckling mice were intracerebrally injected with 1 × 10^5^ PFU of DENV alone, served as a positive control, or co-administered 1 mg/kg of sofalcone on days 1, 3, and 5 post-infection (dpi). Heat-inactivated DENV (iDENV) was used as a negative control. Mice were monitored daily for body weight, clinical score, and survival rate over the 6 days following DENV and sofalcone administration. Viral titers in the brain were determined by plaque assay at 6 dpi or on the day of death for the DENV-infected mice. As shown in Figure 2A, the DENV-injected group exhibited significant body weight loss, approximately 60%, from days 4 to 6 post-infection compared to the iDENV-injected group. In contrast, the sofalcone-treated group lost only 20% of its body weight by 6 dpi. Clinical observations indicated that the sofalcone-treated mice displayed only mild asthenia and paralysis (Figure 2B). As anticipated, sofalcone treatment (1 mg/kg) delayed the lethality of DENV infection, resulting in an 80% survival rate at 6 dpi, compared to the untreated mice (Figure 2C). Additionally, viral titers in the brain tissue of sofalcone-treated mice showed a 2-log reduction compared to untreated DENV-infected mice (Figure 2D).

### 2.2. Sofalcone Suppresses DNV Replication Through the Elevation of HO-1 Expression

Sofalcone has been reported to activate Nrf2-mediated HO-1 expression [17]. Moreover, the induction of HO-1 has been shown to inhibit DENV replication by targeting viral protease activity [13]. To determine if the anti-DENV effects of sofalcone are linked to its ability to activate HO-1 expression, we first examined the effect of sofalcone on HO-1 transcriptional activity using a reporter assay based on the HO-1 promoter. In this assay, Huh-7 cells were transfected with the pHO-1-FLuc reporter vector, followed by DENV infection and treatment with increasing concentrations of sofalcone. The results showed that sofalcone induced DENV-suppressed HO-1 promoter activity in a dose-dependent manner (Figure 3A). To further confirm these findings, RT-qPCR and Western blotting analyses were performed, which revealed that sofalcone increased the levels of HO-1 RNA and protein in a concentration-dependent manner during DENV infection (Figure 3B,C). To validate the role of HO-1 induction in the antiviral activity of sofalcone, DENV-infected Huh-7 cells were exposed to 30 μM sofalcone with or without increasing concentrations (1–15 μM) of the specific HO-1 inhibitor, stannous mesoporphyrin (SnMP), for 3 days. As shown in Figure 4A,B, the suppression of DENV protein and RNA levels by sofalcone was progressively reversed by SnMP treatment. HO-1 gene silencing also reversed the anti-DENV activity of sofalcone (Figure 4C).

### 2.3. Sofalcone Exerts an Antiviral Effect Through Nrf2-Mediated HO-1 Expression

Three transcription factors, Bach1, Keap1, and Nrf2, regulate HO-1 expression through competitive binding to the ARE sequence in the promoter region [18]. To determine whether sofalcone’s induction of HO-1 against DENV replication is dependent on ARE transactivation, we first performed an ARE-driven reporter assay. Huh-7 cells transfected with the p3xARE-Luc were treated with increasing concentrations of sofalcone in the presence of DENV infection for 3 days. As shown in Figure 5A, sofalcone significantly enhanced ARE-mediated luciferase activity dose-dependently. Next, we investigated the protein expression levels of the key transcription factors involved in HO-1 regulation to identify which one was influenced by sofalcone. Western blotting analysis revealed that sofalcone caused a gradual increase in Nrf2 protein levels, with no significant changes observed in the protein levels of Keap1 and Bach1 (Figure 5B). Additionally, sofalcone promoted the nuclear accumulation of Nrf2 protein in a concentration-dependent manner (Figure 5C). To verify that sofalcone’s anti-DENV activity depends on Nrf2-mediated HO-1 induction, DENV-infected Huh-7 cells were subjected to Nrf2 knockdown by transfection of increasing concentrations of Nrf2 shRNA and treated with 30 μM sofalcone for 3 days. DENV RNA replication was assessed by RT-qPCR. As shown in Figure 5D, the reduction in DENV RNA caused by sofalcone was progressively restored following Nrf2 knockdown in a concentration-dependent manner. Together, these results highlight sofalcone’s ability to modulate Nrf2-mediated antioxidant response, supporting its potential as a therapeutic agent in combating viral infections.

### 2.4. Sofalcone Inhibits DENV NS3 Protease Activity and Elevated Antiviral IFN Response

Earlier research demonstrated that biliverdin, a metabolite of HO-1, has been shown to inhibit DENV NS2B/NS3 protease activity, which is associated with enhanced antiviral IFN responses against DENV infection [13]. To explore whether sofalcone’s anti-DENV activity relies on the HO-1-mediated antiviral pathway, we first conducted a cell-based reporter assay to evaluate the effect of sofalcone on anti-DENV NS2B/NS3 protease activity. Briefly, cotransfection of Huh-7 cells was performed using the pCMV-NS2B-NS3 protease expression vector with the reporter vector pEG(∆4B/5)sNLuc [13]. The transfected cells were then treated with sofalcone (5 to 30 μM) for 3 days, followed by luciferase activity measurement. To confirm the role of HO-1 induction in DNEV NS2B/NS3 protease inhibition, combination treatment with 20 μM of the HO-1 inhibitor SnMP was performed. As shown in Figure 6, sofalcone significantly suppressed DENV protease activity (black columns). However, its antiprotease activity was diminished when SnMP was applied (white columns). Based on the inhibition of NS2B/NS3 protease activity, we further examined whether sofalcone’s blockage of the innate immune response to virus infection could restored or stimulated. We first analyzed the expression of IFN genes, including IFN-α-2 and IFN-α-17, in DENV-infected Huh-7 cells treated with sofalcone, both in the presence or absence of SnMP. RT-qPCR results showed sofalcone significantly increased the RNA levels of IFN-α-2 and IFN-α-17, but this effect was attenuated when SnMP was present (Figure 7A). Next, we conducted an IFN-stimulatory response element (ISRE) reporter assay to further confirm that the elevated IFN gene expression was linked to an increased IFN-α protein response. Following transfection with the pISRE-Luc reporter plasmid, Huh-7 cells were infected with DENV and co-treated with sofalcone and SnMP. Luciferase reporter assay results indicated that sofalcone induced a concentration-dependent IFN response, which was reduced by SnMP treatment (Figure 7B). Moreover, sofalcone treatment significantly increased the expression of interferon-stimulated genes (ISGs), including 2′-5′-oligoadenylate synthetase 1 (OAS1), OAS2, and OAS3 (Figure 7C–E). This elevated expression of ISGs was diminished by SnMP. These findings indicate that sofalcone restores the antiviral activity of IFN against DENV replication through Nrf2–HO-1-mediated inhibition of viral protease.

## 3. Discussion

Reactive oxygen species (ROS) generated during DENV infection play both beneficial and detrimental roles in the innate immune system. At moderate levels, ROS interferes with viral infection by activating innate immune responses, but excessive ROS induces high oxidative stress, damages host tissues, and exacerbates inflammatory responses [19,20]. These effects have been proposed to contribute to the progression of DHF and DSS in DENV-infected patients [21]. Numerous studies have demonstrated a correlation between the severity of dengue illness and the oxidative stress response [22]. As a result, antioxidant therapy has been proposed as a potential approach to mitigate disease progression. Sofalcone is clinically used in the treatment of gastrointestinal disorders such as gastritis and peptic ulcers, where its antioxidant capabilities help protect the gastric mucosa [15]. This study found that sofalcone significantly reduced DENV replication and increased the expression of the antioxidant gene heme oxygenase-1 (HO-1) following DENV infection (Figure 1 and Figure 2). HO-1 is a critical enzyme in the oxidative stress response for its cytoprotective, anti-inflammatory, and antioxidant effects [23]. Therefore, sofalcone may serve as a promising therapeutic agent against DENV diseases by enhancing antioxidant defenses.

Previous research has highlighted the crucial role of HO-1 in modulating innate and adaptive immune responses under various conditions, including autoimmune diseases, sepsis, transplantation, and oxidative stress. HO-1 is also recognized as a protective mechanism against oxidative stress induced by viral infections, contributing to the maintenance of tissue homeostasis [24]. Numerous studies have reported that the upregulation of HO-1 is linked to significant antiviral effects against a variety of viruses, including HIV, HBV, HCV, enterovirus 71, influenza virus, respiratory syncytial virus, Ebola virus, and DENV [11]. Like other viral pathogens, DENV employs various strategies to evade innate immune responses by targeting the HO-1-Nrf2 pathway, thus enhancing its infectious potential. The present study reveals that sofalcone significantly inhibits DENV replication through an Nrf2–HO-1-dependent mechanism (Figure 3, Figure 4 and Figure 5). Previous research has demonstrated that antioxidant compounds such as andrographolide and lucidone can activate HO-1, thereby effectively inhibiting DENV replication and reducing DENV-induced vascular leakage in both *in vitro* and *in vivo* models [9]. The Nrf2–HO-1 axis represents a common pathway exploited or disrupted by various viral pathogens to facilitate infection and pathogenesis [8,25,26]. Recent studies have demonstrated that activating the Nrf2–HO-1 axis can reduce virus-induced oxidative stress and cytokine storms by enhancing antioxidant defenses and eliminating excessive inflammatory responses, respectively [27,28]. These pathogenic mechanisms result in tissue damage and organ failure, which are risk factors of disease severity and mortality in viral infection [29]. Therefore, targeting the Nrf2–HO-1 axis can provide potential protective effects against virus-induced tissue injury. Additionally, by targeting this host-dependent pathway rather than virus-specific proteins, such as viral protease and polymerase, we can potentially develop broad-spectrum antivirals that overcome the limitations of virus-specific approaches, such as viral mutation and resistance, potentially addressing the urgent need for effective therapies against emerging pathogens, such as COVID-19 [11,30].

Several nonstructural proteins of DENV, such as NS2A, NS2B/3, NS4A, NS4B, and NS5, have been reported to inhibit IFN production through multiple signaling mechanisms. Notably, the NS2B/3 protease specifically targets STING for proteolytic cleavage, resulting in reduced IRF3 phosphorylation and subsequent inhibition of IFN production in DENV-infected cells [31]. Our findings revealed that sofalcone effectively inhibits DENV replication through inhibition of NS2B/NS3 protease activity and activation of the HO-1-mediated antiviral IFN signaling pathway (Figure 6 and Figure 7). This suggests a protective mechanism in which sofalcone induces HO-1 expression to increase biliverdin levels, which reportedly interfere with DENV NS2B/NS3 protease activity [13], thereby restoring antiviral IFN signaling disrupted by DENV. Recent analyses indicate that the DENV NS2B3 protease targets Nrf2 for lysosomal degradation, elevating ROS levels and thereby counteracting the antiviral responses of IFN, ultimately enhancing viral replication and inflammation. Consequently, sofalcone-induced Nrf2 accumulation thus contributes to synergistic antiviral effects through mitigation of DENV-induced stress and restoration of IFN-mediated antiviral responses via Nrf2-driven antioxidant pathways. High viral loads and circulating DENV NS1 antigen levels are critical factors in the progression of DENV infection from a mild febrile state to a severe hemorrhagic fever/dengue shock syndrome (DHF/DSS). These conditions are associated with intracellular ROS production and an inflammatory storm. Sofalcone, known for its gastric mucosa protective properties, has also been explored for its potential vascular protective effects due to its antioxidant and anti-inflammatory properties. These effects are mediated by an increase in vascular endothelial growth factor production through the Nrf2–HO-1-dependent pathway in gastric epithelial cells [15]. Furthermore, sofalcone inhibits the inflammatory interactions between macrophages and adipocytes, promoting vascular health by reducing inflammation [14], and potentially preventing the cytokine storm often observed in severe dengue cases. Further research is necessary to thoroughly assess the protective effects of sofalcone on DENV-induced plasma leakage using an AG129 mouse model [32]. Additionally, exploring the potential of sofalcone in combination with other antiviral strategies could provide a comprehensive approach to managing DENV infection and potentially other emerging viral pathogens.

Recent decades have brought successful development of direct-acting antivirals (DAAs) targeting viral protease or polymerase for antiviral treatment, such as HCV [33] and COVID-19 [34]. However, a major challenge in developing DAAs is the low fidelity of RNA viral polymerase and the virus’s high replication rate [35], which leads to mutations in the viral genome associated with drug resistance [36]. Host immune responses, such as the activation of interferons (IFNs), offer potential protective effects against DENV progression. Our research highlights that sofalcone effectively inhibits DENV through inhibition of NS3 protease activity and subsequent activation of antiviral IFN responses mediated by the Nrf2–HO-1 signaling pathway, as shown in Figure 8. Moreover, sofalcone also provides protection against lethality and illness caused by DENV infection in a mouse model (Figure 2), offering a comprehensive approach to managing DENV infection. Importantly, sofalcone is already approved for clinical use in the treatment of gastritis and peptic ulcers, meaning its safety and pharmacokinetics are well established [37]. Due to already passing several regulatory hurdles, sofalcone can accelerate repurposing as an antiviral drug. However, further clinical studies are necessary to validate these hypotheses, evaluate the clinical efficacy of sofalcone, and determine its safety profile at effective doses in managing DENV infection.

## 4. Materials and Methods

### 4.1. Cell Culture and Reagents

The human hepatoma cell line Huh-7 was passaged in Dulbecco’s modified Eagle’s medium containing 10% heat-inactivated fetal bovine serum, 1% antibiotic–antimycotic, and 1% nonessential amino acids. The cells were incubated at 37 °C in a humidified atmosphere with 5% CO_2_. Sofalcone and tin mesoporphyrin (SnMP), both obtained from Sigma Aldrich Co. (St. Louis, MO, USA), were utilized in the experiments. Stock solutions of all compounds were prepared at a concentration of 100 mM in 100% dimethyl sulfoxide (DMSO), with the final DMSO concentration in all experimental maximum concentration maintained at 0.1%.

### 4.2. DENV-Infected Cell Model

Huh-7 cells were seeded into 24-well plates at a density of 2 × 10^5^ cells/mL and incubated overnight at 37 °C. When the cell monolayer reached approximately 80% confluence, DENV serotype 2 (DENV-2; 16681 strain) was introduced at a multiplicity of infection (MOI) of 0.2. After a 2 h incubation period, the culture medium was replaced with DMEM supplemented with 10% FBS, along with the tested compound. Following a 3-day incubation, cell lysates or total RNA was harvested for analysis using Western blotting or RT-qPCR. DENVs of different serotypes (DENV-1: DN8700828; DENV-3: DN8700829A; DENV-4: S9201818) were obtained from the Centers for Disease Control, Department of Health, Taiwan.

### 4.3. Animal Ethics and Virus Inoculation Procedures for ICR Suckling Mice

This study utilized six-day-old ICR mice obtained from BioLASCO in Taipei, Taiwan for intracranial inoculation with DENV-2 (PL046 strain) at a concentration of 2.5 × 10^5^ pfu/mouse. The mice were inoculated with either active or heat-inactivated DENV-2 medium supplemented with 2% FBS. Subsequently, the mice were intracranially injected with sofalcone or saline at days 2 and 4. Experimental mice were subjected to daily monitoring over a 6 to 10 day period. Assessments included body weight measurements, clinical sign evaluation, and mortality documentation over 6 days. Clinical severity was evaluated according to the following criteria: 0: healthy; 1: weight loss, ruffled hair, or hunchbacked appearance; 2: reduced mobility; 3: moving with difficulty and limb weakness; 4: limb paralysis; and 5: moribund or death. In the experiment, mice with body weight below 25% of their original weight were considered dead and their brain tissue was harvested for viral titer measurement following humane euthanasia immediately, while others were euthanized and their brain tissue was harvested for viral titer measurement on day 6. The experiment was approved by the Animal Care and Use Committee of Kaohsiung Medical University (IACUC, 109125) and conducted in full compliance with the US National Institutes of Health Guide for the Care and Use of Laboratory Animals. Mice were maintained at the Animal Facility of Kaohsiung Medical University, and all experimental procedures followed the animal experimentation guidelines established by Taiwan’s National Science and Technology Council.

### 4.4. Plaque Assay

BHK-21 cells were seeded at of 9 × 10^4^ cells per well and infected with viral particles isolated from DENV-infected mouse brains. The cells were exposed to serial dilutions of the virus for 2 h, after which the medium was replaced with DMEM supplemented with 2% FBS and 0.8% methylcellulose (Sigma-Aldrich, St. Louis, MO, USA). Cells were incubated for 4 days, then fixed and stained by adding a solution containing 1% crystal violet, 0.64% NaCl, and 2% formalin and incubated for 1 h at room temperature. Following tap water washing, plaques were counted to determine viral titer.

### 4.5. Preparation of Nuclear Fraction and Western Blot Analysis

Nuclear fractions were isolated using a cytoplasmic and nuclear protein extraction kit (BIOTOOLS, New Taipei City, Taiwan) according to the manufacturer’s instructions. Briefly, following cell harvest and PBS washing, cells were pelleted by centrifugation at 16,000×g for 5 min. Cell pellets were resuspended in cytoplasmic extraction buffer with protease inhibitors and incubated on ice, then centrifuged again at 16,000× *g* for 5 min. the cytoplasmic fraction (supernatant) was collected, and the nuclear pellet was resuspended in nuclear extraction buffer for Western blotting analysis. Cell lysates were separated on 10% SDS-PAGE and transferred onto a polyvinylidene difluoride membrane (Pall Fluoro Tran^®^ W) provided by PALL Life Sciences (Melbourne, VIC, Australia). The membranes were blocked with 5% skim milk and incubated separately with anti-DENV NS2B (1:3000; GeneTex, Irvine, CA, USA), anti-HO-1 (1:2000; GeneTex), anti-Nrf2 (1:2000; GeneTex), anti-Keep1 (1:2000; Abcam, Cambridge, MA, USA), and anti-Bach1 (1:2000; Abcam), in which blots were probed with GAPDH (1:5000; GeneTex) antibody as a protein loading control. Following incubation with horseradish peroxidase (HRP)-labeled goat anti-rabbit or goat anti-mouse antibodies (1:10,000; GeneTex), the bands were visualized using an enhanced chemiluminescence (ECL) detection system, following the manufacturer’s instructions (PerkinElmer, Shelton, CT, USA).

### 4.6. Cell Viability Assessment

Cell viability was assessed using the CellTiter 96^®^ AQ_ueous_ One Solution cell Proliferation Assay (MTS) kit (Promega Corporation, Madison, WI, USA). In 96-well plates, cells were seeded at a density of 5 × 10^3^ per well and treated with increasing concentrations of sofalcone for 3 days. Post-treatment, culture supernatant was removed, and 100 μL of phenol red-free medium containing 20 μL of MTS reagent was added to each well for viability assessment. The cells were then incubated at 37 °C for 4 h. Relative cell viability was determined by measuring absorption at 490 nm using a Synergy HTX multimode reader (BioTek, Winooski, VT, USA).

### 4.7. DENV NS2B/NS3 Protease Assay

The DENV NS2B/NS3 protease activity was evaluated using established protocols [13]. Huh-7 cells were cotransfected with 0.25 μg of the NS2B/NS3 protease reporter vector pEG(∆4B/5)sNLuc and 0.75 μg of either the active NS2B/NS3 protease vector pNS2B-NS3 or the catalytically inactive mutant NS2N/NS3 protease vector pCMV-NS2B-NS3(D75A/S135A). After transfection, the cells were treated with varying concentrations of sofalcone for 3 days. The supernatants were then collected for nanoluciferase activity measurements using the Nano-Glo^®^ Luciferase Assay System (Promega). Each transfection mixture included 0.1 μg of the firefly luciferase expression vector (pCMV-FLuc) as a normalization control, relative to the nanoluciferase activity. 

### 4.8. Real-Time Quantitative Reverse Transcription Polymerase Chain Reaction (RT-qPCR)

RNA extraction was performed using the total RNA minipump purification kit (GMbiolab Co., Ltd., Taipei, Taiwan), and cDNA synthesis was carried out with M-MLV reverse transcriptase (Promega, WI, USA) according to the manufacturer’s protocol. RT-qPCR was used to quantify the expression levels of DENV NS5B, HO-1, and the OAS family (OAS1-2) using the primer sequences listed in Table 1. RNA levels in each sample were normalized to GAPDH mRNA levels as an internal control and quantified using the ABI Step One Real-Time PCR System following the manufacturer’s protocols.

### 4.9. Transient Transfection and Luciferase Reporter Assay

Plasmid transfection was performed using T-Pro™ reagent (Ji-Feng Biotechnology Co. Ltd., Taipei, Taiwan), following the standard manufacturer’s protocol. Huh-7 cells were plated at a density of 5 × 10^4^ per well and subsequently transfected with each reporter plasmid containing specific transcriptional response elements linked to firefly luciferase: pHO-1-Luc (human HO-1 promoter), p3xARE-Luc (three Nrf2-dependent antioxidant response elements), or pISRE-Luc (IFN-stimulated response element) to monitor HO-1, Nrf2, or IFN responses, respectively. Following transfection, cells were infected with DENV-2 and treated with sofalcone at the concentrations indicated. After a 3-day incubation period, cell lysates were prepared and subjected to luciferase activity analysis using the Bright-Glo Luciferase assay system (Promega Corporation, Madison, WI, USA). For transfection efficiency correction, 0.1 μg of secreted alkaline phosphatase (SEAP) expression vector (pCMV-SEAP) was cotransfected, and luciferase reporter activities were normalized to SEAP activity levels. HO-1 (NM_002133), Nrf2 (NM_006164), and EGFP Small hairpin RNAs (shRNAs) were obtained from the National RNAi Core Facility, Institute of Molecular Biology/Genomic Research Center, Academia Sinica, Taiwan, and sequencing verification was performed on all DNA fragments.

### 4.10. Statistical Analysis

Data were analyzed by one-way analysis of variance (ANOVA) test, followed by Tukey’s test for comparison of multiple groups using GraphPad Prism 6.0 (GraphPad Software, LA Jolla, CA, USA) and are presented as the mean ± standard deviation (SD), based on at least three independent experiments performed in triplicate. Statistical significance was defined as a ** p* <0.05 or *** p* < 0.01.

## 5. Conclusions

In conclusion, this study demonstrates that sofalcone, a clinically approved gastroprotective agent, exhibits potent antiviral activity against all four serotypes of DENV through activation of the Nrf2–HO-1–IFN pathway. In DENV-infected ICR suckling mice, sofalcone treatment improved survival rate and reduced viral titers in the brain tissue, confirming its protective efficacy. The underlying mechanism involves HO-1 upregulation via Nrf2 activation, resulting in inhibition of viral protease activity and stimulation of antiviral genes. These findings suggest sofalcone’s potential as an anti-DENV therapeutic and highlight the therapeutic value of targeting the Nrf2–HO-1 axis to enhance host antiviral responses while providing potential protection against virus-induced tissue damage through antioxidant and anti-inflammatory effects. Clinical studies are needed to validate these results and explore broader antiviral applications.

## Figures and Tables

**Figure 1 ijms-26-05921-f001:**
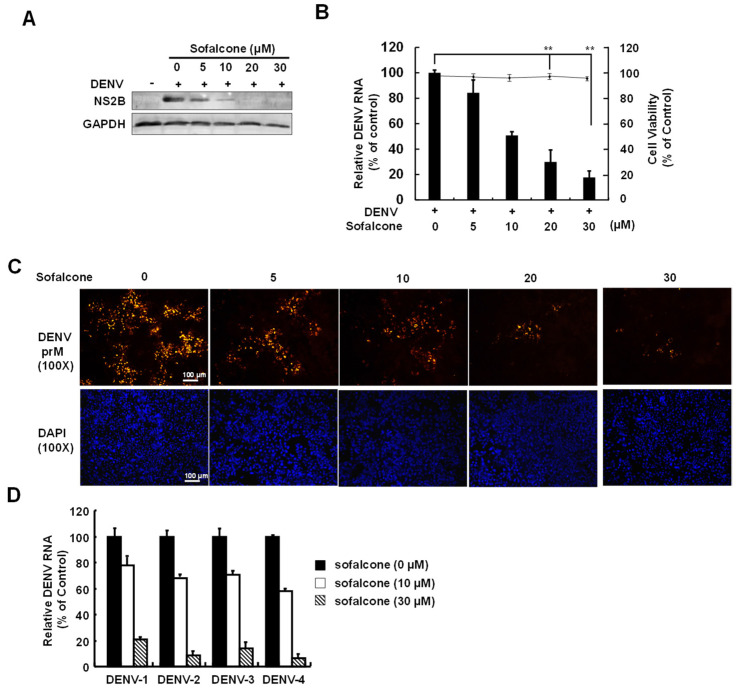
The efficacy of sofalcone in inhibiting DENV protein synthesis and RNA replication. A range of sofalcone concentrations (5, 10, 20, and 30 µM) was administered over 3 days to Huh-7 cells inoculated with DENV at an MOI of 0.2. The cell lysate was then collected and analyzed using (**A**) Western blot with DENV NS2B antibody and (**B**) RT-qPCR with the DENV NS5B primer to measure viral protein and RNA levels, respectively. Simultaneously, cellular toxicity was assessed using the MTS assay. In the Western blot analysis, GAPDH was used as the loading control, while viral RNA levels were normalized to cellular *gapdh* mRNA levels in the RT-qPCR analysis. Bars display the relative levels of DENV RNA, and the line graph indicates cell viability at the indicated concentrations of sofalcone. (**C**) Following treatment with the same concentrations, fluorescence immunostaining was performed on DENV-infected Huh-7 cells using anti-dengue prM antibodies and DAPI staining. (**D**) Sofalcone treatment effectively inhibited all four DENV serotypes: 1, 2, 3, and 4. Huh-7 cells were infected with the respective DENV serotypes (DENV-1: DN8700828; DENV-2: 16681; DENV-3: DN8700829A; DENV-4: S9201818) at a multiplicity of infection (MOI) of 0.2 and treated with varying concentrations of sofalcone (10 to 30 μg/mL) for 3 days. The relative levels of DENV RNA and cell viability are expressed as percentage changes compared to sofalcone-untreated cells with 0.1% DMSO, designated as 100%. The data are presented as the mean ± standard deviation (SD) from three independent experiments. Statistically significant differences are denoted by ** *p* < 0.01.

**Figure 2 ijms-26-05921-f002:**
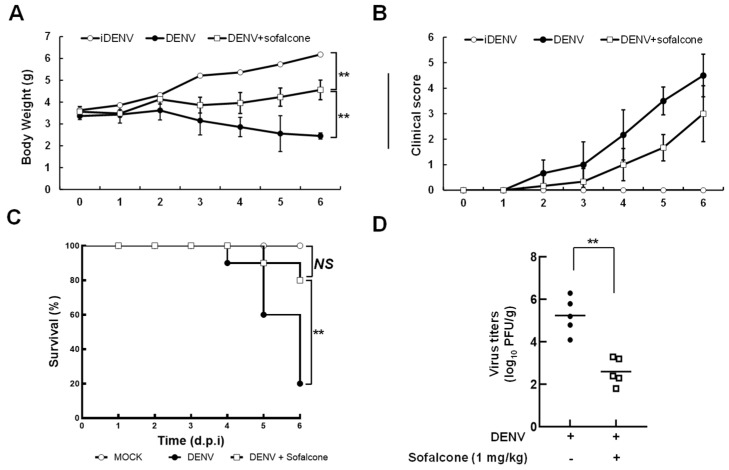
Protective effect of sofalcone on ICR suckling mice against lethal DENV infection. Six-day-old ICR-suckling mice were divided into three groups. One group received intracranial inoculation with DENV at a dose of 2 × 10^5^ pfu per mouse, followed by sofalcone treatment at a concentration of 1mg/kg or saline treatment on days 2 and 4 by intracranial inoculation. Another group was administered 60 °C heat-inactivated DENV (iDENV) as a negative control. From day 1 to 6, we measured (**A**) body weight, (**B**) clinical score, (**C**) survival rate, and (**D**) virus titers in the mouse brain using a plaque assay. Clinical symptom assessment and viral titer measurement timing are detailed in the Section 4. Each datapoint reflects the mean values for all animals in the group (n = 5). Statistical significance is indicated as ** *p* < 0.01.

**Figure 3 ijms-26-05921-f003:**
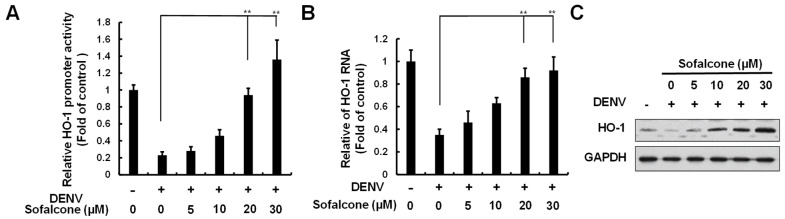
Induction of HO-1 expression by sofalcone in DENV-infected Huh-7 cells. Sofalcone treatment enhanced the HO-1 expression reduced by DENV at the (**A**) promoter, (**B**) RNA levels, and (**C**) protein levels. Huh-7 cells infected with DENV (MOI of 0.2) were transfected with the HO-1 promoter-driven firefly luciferase reporter plasmid pHO-1-FLuc and subsequently treated with sofalcone at concentrations ranging from 5 to 30 μM for 3 days. Luciferase assay was measured to assess promoter activity. HO-1 RNA and protein levels were quantified using RT-qPCR and Western blotting, respectively. The relative levels of HO-1 RNA were normalized to cellular *gapdh* mRNA. GAPDH was used as a loading control for both RT-qPCR and Western blot analyses. Both HO-1 promoter activity and RNA levels were expressed as fold changes relative to untreated parental Huh-7 cells, which were set to 1. The experiment was conducted in replicates, and error bars represent SD. Statistical significance is indicated by and ** *p* < 0.01.

**Figure 4 ijms-26-05921-f004:**
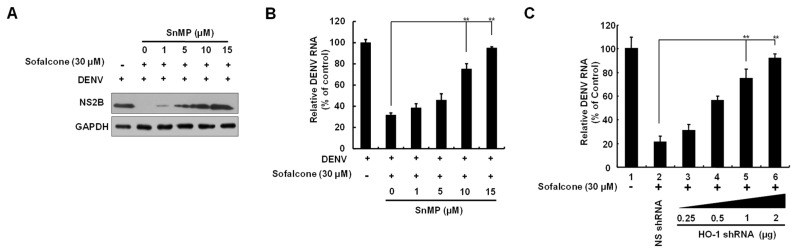
Reduction in the antiviral efficacy of sofalcone by HO-1 inhibitor SnMP and HO-1 knockdown. Huh-7 cells infected with DENV (MOI of 0.2) were co-treated with sofalcone (30 μM) and varying concentrations of the HO-1 inhibitor SnMP, ranging from 2.5 to 20 μM, or transfection of HO-1 shRNA expression vector, ranging from 0.25 to 2 μg, for 3 days. (**A**) Viral protein levels and (**B**,**C**) RNA levels were assessed using Western blotting and RT-qPCR, respectively. GAPDH was used as a loading control for both RT-qPCR and Western blot analyses. Transfection of non-specific shRNA expression vector (named NS shRNA; shEGFP) served as a negative control. Viral RNA levels were expressed as percentage changes relative to DENV-infected Huh-7 cells treated with 0.1% DMSO, which was designated as 100%. This experiment was conducted in triplicate. Error bars represent SD. Statistical significance is indicated by ** *p* < 0.01.

**Figure 5 ijms-26-05921-f005:**
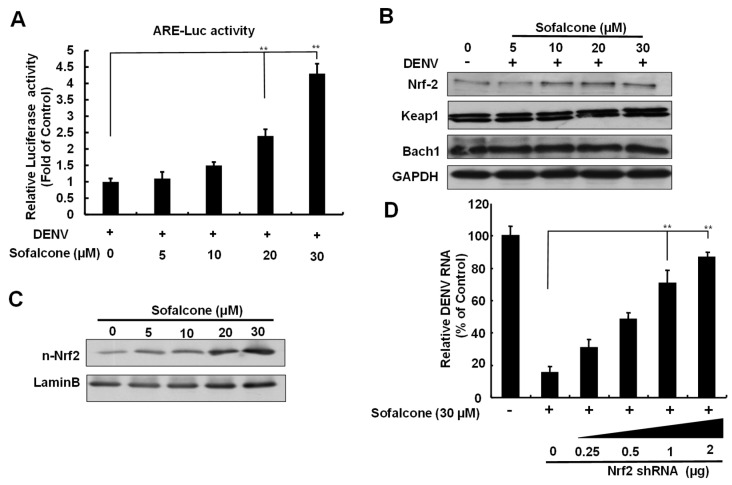
Induction of HO-1 expression against DENV by sofalcone through Nrf2 induction. (**A**) Huh-7 cells infected with DENV (MOI of 0.2) were transfected with the p3xARE-Luc reporter plasmid and subsequently treated with 5 to 30 μM sofalcone for 3 days. Luciferase activity was measured in total cell lysates to assess Nrf2-mediated transcriptional activity, expressed as fold changes relative to the control treatment with 0.1% DMSO defined as 1. (**B**,**C**) Western blotting was performed using anti-Bach1, anti-Keap1, anti-Nrf2, and anti-GAPDH antibodies to evaluate the protein levels of key regulators involved in the Nrf2 pathway, with a focus on nuclear accumulation of Nrf2 in the presence of DENV infection and sofalcone treatment. GAPDH and the nuclear marker larmin B were used to confirm equal loading of the cell lysates. (**D**) The impact of Nrf2 knockdown on the antiviral activity of sofalcone. RNA levels were determined by RT-qPCR in the presence of Nrf2 shRNA expression or non-specific shRNA expression (named: NS shRNA; shEGFP) under sofalcone (30 μM) treatment. Viral RNA levels were calculated as percentage changes using DENV-infected Huh-7 cells treated with 0.1% DMSO, defined as 100%. The experiment was conducted in replicates, and error bars represent SD. Statistical significance is indicated as ** *p* < 0.01.

**Figure 6 ijms-26-05921-f006:**
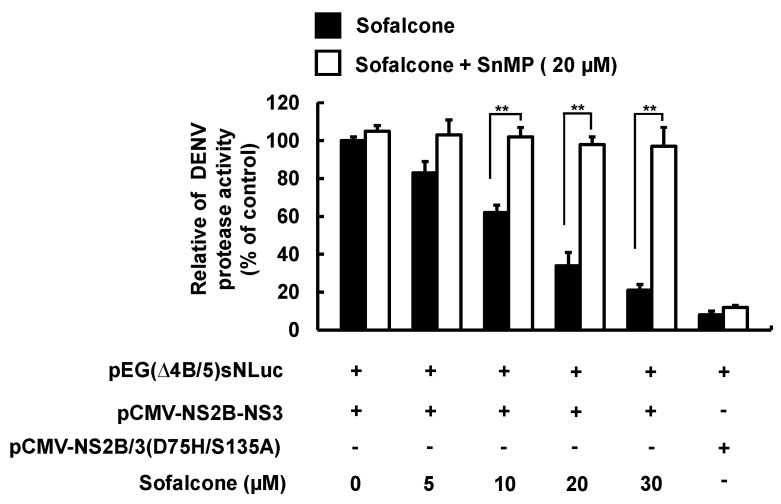
Inhibitory effect of sofalcone on DENV protease activity. Huh-7 cells were cotransfected with DENV NS2B/3 protease expression vector pCMV-NS2B-NS3 and protease reporter vector pEG(Δ4B/5)sNLuc, in which transfection of pCMV-NS2B-NS3(D75A/S135A), a NS3 protease mutant, served as a negative control. The cells were then treated with varying concentrations of sofalcone, with or without adding 20 μM SnMP. Protease activity was assessed using a luciferase activity assay. This experiment was independently replicated three times, and the results are presented with SD indicated by the error bars. Statistical significance is indicated as ** *p* < 0.01.

**Figure 7 ijms-26-05921-f007:**
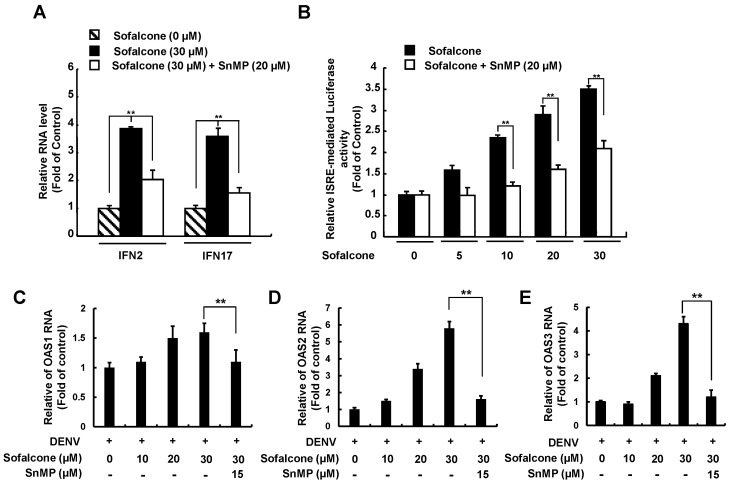
Induction of IFN-mediated antiviral responses by sofalcone through HO-1 activation. (**A**) Sofalcone enhanced IFN gene expression, an effect that was diminished by SnMP treatment. DENV-infected Huh-7 cells were treated with 30 μM sofalcone alone or in combination with 20 μM SnMP for 3 days. Following total RNA extraction, IFN-2 and IFN-17 RNA levels were measured using RT-qPCR. (**B**) Sofalcone dose-dependently elevated the IFN response, but this effect was diminished by SnMP treatment. DENV-infected Huh-7 cells transfected with the IFN response reporter vector pISRE-Luc were exposed to 5–30 μM sofalcone with or without 20 μM SnMP for 3 days. Total cell lysates were analyzed for luciferase activity. (**C**–**E**) Sofalcone gradually upregulated IFN-mediated antiviral gene expression but was diminished with SnMP treatment. Total RNA of DENV-infected Huh-7 cells with or without adding sofalcone and SnMP treatment was extracted to quantify OAS1, OAS2, and OAS3 RNA levels using RT-qPCR. Relative RNA levels were normalized to cellular *gapdh* mRNA levels. “0” indicates treatment with 0.1% DMSO. The RNA levels and ISRE response activity are normalized to the DMSO control and presented as fold changes (DMSO =1). This experiment was conducted in triplicate. Error bars represent SD. Statistical significance is indicated by ** *p* < 0.01.

**Figure 8 ijms-26-05921-f008:**
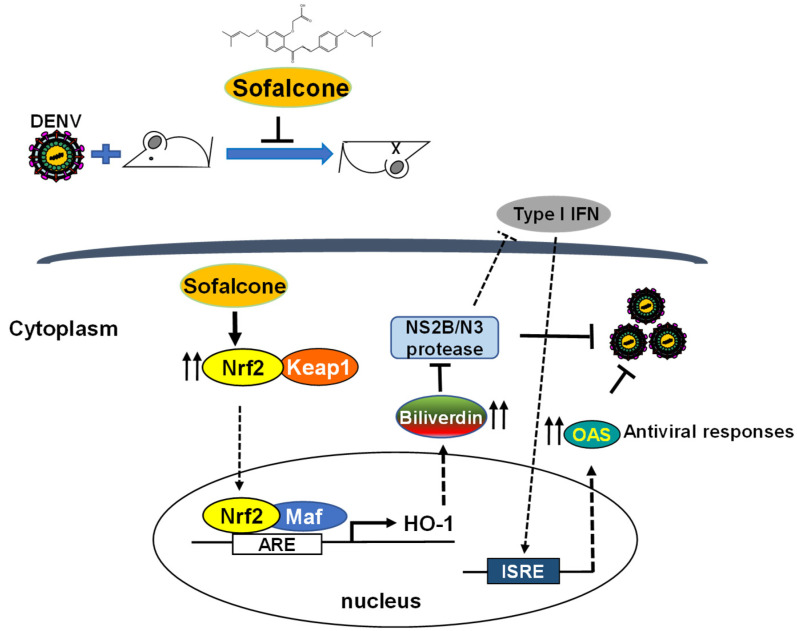
Mechanism by which sofalcone inhibits DENV replication. Sofalcone suppresses DENV replication by triggering Nrf2–HO-1 signaling, which inhibits NS2B/3 protease activity and enhances antiviral interferon response. In vivo experiments confirm the protective efficacy of sofalcone against lethal DENV infection in ICR suckling mice.

**Table 1 ijms-26-05921-t001:** Oligonucleotide sequences for real-time RT-PCR.

Oligonucleotide Name	Sequence 5′-3′
3′NS5B	5′-GGAAACCA GCTGCCCATCA
3′NS5B	5′-CCTCCACGGATAGAAGTTTAA
5′GAPDH	5′-GTCTTCACCACCATGGAGAA
3′GAPDH	5′-ATGGCATGGACTGTGGTCAT
5′IFN-2	5′-GCAAGTCAAGCTGCTCTGTG
3′IFN-2	5′-GATGGTTTCAGCCTTTTGGA
5′IFN-17	5′-AGGAGTTTGATGGCAACCAG
3′IFN-17	5′-CATCAGGGGAGTCTCTTCCA
5′OAS1	5′-CAAGCTTAAGAGCCTCATCC
3′OAS1	5′-TGGGCTGTGTTGAAATGTGT
5′OAS2	5′-ACAGCTGAAAGCCTTTTGGA
3′OAS2	5′-GCATTAAAGGCAGGA AGCAC
5′OAS3	5′-CACTGACATCCCAGACGATG
3′OAS3	5′-GATCAGGCTCTTCAGCTTGG

## Data Availability

Data are unavailable due to privacy.

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
