# Peer review of "Sofalcone Suppresses Dengue Virus Replication by Activating Heme Oxygenase-1-Mediated Antiviral Interferon Responses"

_ijms, 2025, doi:10.3390/ijms26135921_

Round 1

Reviewer 1 Report

Comments and Suggestions for Authors

Current study by Ou et al. assessed the anti-DENV effects of sofalcone. The authors show sofalcone upregulate Nrf2-mediated HO-1 expression. The study shows that sofalcone can inhibit DENV NS2B/NS3 protease activity, potentially restoring or amplifying antiviral IFN responses and synergistically suppressing viral replication. Further, protective effects of sofalcone were examined in vivo using the DENV-infected ICR-suckling mouse model.

Figure 1B: Bars depict relative DENV RNA while line graph show cell viability with indicated Sofalcone concentrations.

Line 315: Stock solutions of all compounds

Line 317: maximum concentration maintained at 0.1%.

Line 342: DNEV> DENV

Reviewer 2 Report

Comments and Suggestions for Authors The subject of the manuscript is interesting and contemporary. Dengue disease is expanding its
range to the temperate zone as a result of the climate changes that have occurred.
The possibility
that the disease can be severe and progress to a fatal outcome necessitates the search for
effective antiviral agents.
The authors prove in in vitro and in vivo experiments
the effectiveness of a new therapeutic agent - sofalconе - with several mechanisms of
antiviral action.
I have no objections regarding the design of the study, the materials and methods used, and the
analysis of the obtained results.
I have some comments regarding the structure of the manuscript. Materials and methods are placed
after the introduction, not at the end of the manuscript after the discussion.
I think the manuscript needs to be restructured.

Reviewer 3 Report

Comments and Suggestions for Authors

Dear Authors,
The manuscript submitted to me for review undoubtedly touches on a topical issue. However, some questions and recommendations arise.

  1. The expression on line 26-"half-maximal effective
    concentration 50%" is not correct for a virological study. It is correct in my opinion- the dose inhibiting 50% of the viral yield compared to the control.
  2. Line 29-what do you mean-"mechanically"....
  3. Line 43 - from what you wrote, it appears that constant treatment leads to a lack of an effective trivalent vaccine. Perhaps you could correct the sentence, since the problem is not in the treatment, but in a number of characteristic features of the virus, its escape from the immune response and the ability to form sufficiently neutralizing antibodies.
  4. I would recommend that you prepare a list of abbreviations from the manuscript for the convenience of the reader.
  5. Active oxygen radicals also have antiviral effects. There are publications on this topic. Please check and clarify.
  6. Line 97-98 - it is not clear whether there is a positive control - i.e. animals infected and untreated with the substance. Please clarify.
  7. It is not clear in the in vivo experiments whether the toxicity of the preparation has been determined.
  8. Line 101 and 102 - please clarify whether in both cases it is e iDENV. If it is quite unclear, why are you comparing two identical groups.
  9. Line 112-please check if it is MTS assay or MTT
  10. Line 264- Do you think that influencing only this pathway is enough to sufficiently deal with the viral infection? From what you have written, it is rather clear that it helps to stop the side effects of the infection. You could review the action of active oxygen radicals generated, for example, by cold plasma and their action. You could enrich the discussion.
  11. Line 303-I think it's pretty bold. If you rely only on interferon and its active synthesis, I think it's not enough. Did you create Figure 8? If not, please indicate the authors.
  12. I don't see a Conclusion section.
